# Vertical transmission of attaching and invasive *E. coli* from the dam to neonatal mice predisposes to more severe colitis following exposure to a colitic insult later in life

Meghan Wymore Brand[1], Alexandra L. Proctor[1], Jesse M. Hostetter[2¤], Naihui Zhou[1], Iddo Friedberg[1], Albert E. Jergens[3], Gregory J. Phillips[1], Michael J. Wannemuehler[1]*

**1** Department of Veterinary Microbiology and Preventive Medicine, College of Veterinary Medicine, Iowa State University, Ames, IA, United States of America, **2** Department of Veterinary Pathology, College of Veterinary Medicine, Iowa State University, Ames, IA, United States of America, **3** Department of Veterinary Clinical Sciences, College of Veterinary Medicine, Iowa State University, Ames, IA, United States of America

¤ Current address: Department of Pathology, College of Veterinary Medicine, University of Georgia, Athens, GA, United States of America

* mjwannem@iastate.edu

**Data Availability Statement:** Data deposited in two separate repositories. R code is deposited at https://github.com/FriedbergLab/neonatal_mouse_

## Abstract

The gastrointestinal microbiota begins to be acquired at birth and continually matures through early adolescence. Despite the relevance for gut health, few studies have evaluated the impact of pathobiont colonization of neonates on the severity of colitis later in life. LF82 is an adherent invasive *E. coli* strain associated with ileal Crohn's disease. The aim of this study was to evaluate the severity of dextran sodium sulfate (DSS)-induced colitis in mice following *E. coli* LF82 colonization. Gnotobiotic mice harboring the altered Schaedler flora (ASF) were used as the model. While *E. coli* LF82 is neither adherent nor invasive, it was been demonstrated that adult ASF mice colonized with *E. coli* LF82 develop more severe DSS-induced colitis compared to control ASF mice treated with DSS. Therefore, we hypothesized that *E. coli* LF82 colonization of neonatal ASF mice would reduce the severity of DSS-induced inflammation compared to adult ASF mice colonized with *E. coli* LF82. To test this hypothesis, adult ASF mice were colonized with *E. coli* LF82 and bred to produce offspring (LF82$_N$) that were vertically colonized with LF82. LF82$_N$ and adult-colonized (LF82$_A$) mice were given 2.0% DSS in drinking water for seven days to trigger colitis. More severe inflammatory lesions were observed in the LF82$_N$ + DSS mice when compared to LF82$_A$ + DSS mice, and were characterized as transmural in most of the LF82$_N$ + DSS mice. Colitis was accompanied by secretion of proinflammatory cytokines (IFNγ, IL-17) and specific mRNA transcripts within the colonic mucosa. Using 16S rRNA gene amplicon sequencing, LF82 colonization did not induce significant changes in the ASF community; however, minimal changes in spatial redistribution by fluorescent in situ hybridization were observed. These results suggest that the age at which mice were colonized with *E. coli* LF82 pathobiont differentially impacted severity of subsequent colitic events.

microbiome. The remaining data sets are deposited at DOI 10.17605/OSF.IO/9KVJR.

**Funding:** The authors acknowledge support from the following organizations: Kenneth Rainin Foundation Innovator (https://krfoundation.org/health/grants/innovatorawards/) awarded to AEJ, MJW, GJP; the National Institute of General Medical Sciences of the NIH (R01GM099537-1A1; http://www.nigms.nih.gov) awarded to GJP, MJW, AEJ, and JMH; the National Cancer Institute of the NIH (R03CA195305; http://www.nci.nih.gov) awarded to GJP and MJW; Directorate for Biological Sciences of the NSF (1551363; http://www.nsf.dbi.gov) awarded to IF. The funders had no role in study design, data collection andanalysis, decision to publish, or preparation of the manuscript.

**Competing interests:** The authors have declared that no competing interests exist.

## Introduction

The human intestine is home to a diverse population of bacteria that reaches $10^{11}$ cells/gram contents in the colon, providing benefits to the host including vitamin production, immune system development, and competitive exclusion of pathogens [1–4]. The gastrointestinal (GI) microbiota is initially acquired in the neonatal period, evolves throughout the postnatal period, and is influenced by diet, antibiotics, and other environmental exposures (e.g., infectious agents) to shape a microbial community that reaches maturity in the young adult host [5, 6].

It has long been thought that in normal circumstances, mammals are born sterile, and the birth process provides the first exposure to microbes, although recent studies challenge that dogma, demonstrating that microbial exposure can occur in utero [7–11]. Additionally, recent studies have demonstrated an impact of the dam's microbial exposures on the mucosal immune responsiveness of offspring [12, 13]. Regardless of when microbes are first introduced into the GI ecosystem, these early microbial exposures (i.e., foundational colonizers) are fundamental to the developing composition of the resident microbial community, establishing GI physiology, and maturation of the mucosal immune system. It has long been recognized that the microbiota impacts the development of mucosal and systemic immunity, as previous studies in germfree mice demonstrated the presence of an underdeveloped mucosal immune system, as well anatomical, histological, and physiological differences compared to conventionally-reared (CONV-R) mice [14–18]. More recent studies have identified distinctly different tissue-specific transcriptional profiles of immune-related genes between mice colonized with a microbiota as neonates compared to being colonized as adults [19, 20].

Multiple studies have demonstrated that early microbial exposures have lasting influence on the host. In the 1960s, Dubos, Savage, and Schaedler introduced the concept of "Biological Freudianism," stating that early environmental factors impact health later in life [21, 22]. These early studies demonstrated differences in weight gain of young mice that were entirely dependent on the composition of their intestinal microbiota [21, 22]. The development of atopy has been particularly well studied in relation to development of the gut microbiota in early life [23–29]. Other studies have demonstrated that altering the microbiota during early life, such as with antibiotics, has a lasting metabolic impact, and may influence the development of chronic inflammatory conditions like inflammatory bowel disease (IBD), obesity, and asthma [30–34].

Inflammatory bowel diseases, including Crohn's disease and ulcerative colitis, are a group of chronic intestinal disorders with a multifactorial etiology consisting of complex interactions between the GI microbiota, genetics, environment, and the immune system [35]. Gastrointestinal pathogens have been implicated as etiological contributors to IBD, notably adherent and invasive *Escherichia coli* (AIEC) in ileal Crohn's disease [36–38]. Though usually diagnosed in young adults (i.e., 2nd to 4th decade of life), pediatric IBD is increasing in worldwide prevalence which suggests a greater role for environmental exposures, as more children are presenting with severe colitis at younger ages [39–41].

One challenge to studying the pathogenesis of AIEC in experimental models is the lack of chronic, stable colonization in immunocompetent, microbiota-harboring, murine models. In humans, AIEC attach to and invade the epithelium via the CEACAM6 receptor [42]. This receptor is not naturally expressed in mice, although studies employing CEACAM6 transgenic mice demonstrate that AIEC strains can persistently colonized and induce intestinal inflammation in CONV-R mice [42, 43]. *E. coli* LF82 is an AIEC isolate recovered from an ileal Crohn's disease patient and in CONV-R mice, *E. coli* LF82 does not readily colonize the gut in the absence of antibiotic pretreatment to facilitate dysbiosis and colonization [37, 44, 45]. In contrast, *E. coli* LF82 has been demonstrated to chronically colonize immunocompetent

gnotobiotic C3H/HeN and C57BL/6 mice harboring the altered Schaedler flora (ASF) in the absence of antimicrobial perturbation [46–48].

We have previously demonstrated that *E. coli* LF82 colonization increases the sensitivity of adult ASF mice to dextran sodium sulfate (DSS)-induced colitis [46]. The aim of the current study was to evaluate changes in the relative abundance and spatial distribution of the ASF and disease severity induced by exposure to DSS following colonization of ASF mice with *E. coli* LF82 as a neonate versus as an adult. We hypothesized that ASF C3H/HeN mice colonized with *E. coli* LF82 as a neonate would develop less severe DSS-induced inflammation compared to ASF mice colonized with *E. coli* LF82 as young adults. We reasoned that *E. coli* LF82 colonization as a neonate with the ASF microbiome would perturb the microbiota less compared to LF82 colonization in young adult ASF mice. However, we demonstrated increased severity of disease associated with DSS-induced colitis in ASF mice colonized with *E. coli* LF82 as neonates. Moreover, the increased severity of colitis was characterized by the presence of transmural inflammation and increased proinflammatory cytokine production and secretion.

## Materials and methods

### Animals

Adult gnotobiotic C3H/HeN mice (originally obtained from Taconic BioSciences) harboring the altered Schaedler flora (ASF) were maintained in Trexler plastic isolators [47, 49]. Fecal or cecal samples from mice in the gnotobiotic colony were evaluated routinely for presence of contaminants by aerobic culture and/or 16s sequencing. ASF members include: ASF356, *Lachnospiraceae* strain.; ASF360, *Lactobacillus intestinalis;* ASF361, *Lactobacillus murinus*; ASF457, *Mucispirillum schaedleri*; ASF492, *Eubacterium plexicaudatum*; ASF500, *Colidextribacter* sp.; ASF502, *Schaedlerella arabinosiphila*; and ASF519, *Parabacteroides goldsteinii*. [47, 49]. Mice were maintained on an irradiated diet (Teklad 2919, Envigo, Huntingdon, Cambridgeshire UK) and autoclaved water for the duration of all experiments. Male and female animals were used in all groups in approximate equal number. All animal experiments were approved by Iowa State University's Institutional Animal Care and Use Committee log number 9-04-5755-M. Procedures were carried out in accordance with the recommendations in the *Guide for the Care and Use of Laboratory Animals* of the National Institutes of Health and all efforts were made to reduce animal discomfort and suffering. Though no animals reached humane endpoints requiring removal from these studies, the mice were closely monitored such that they would have been humanely euthanized in the event that any animal had an adverse response to oral gavage, experienced severe trauma from a cage mate (e.g., excessive barbering, fighting), or demonstrated loss of more than 15% of initial body weight after DSS treatment.

### Experimental design

Mice were orally colonized with *E. coli* LF82 by oral gavage three weeks prior to pairing for breeding. Males were removed after 2.5 weeks to prevent rebreeding and prevent any microbial impact of coprophagy by the pups. Pups born to *E. coli* LF82-infected dams were colonized by vertical transmission during the process of microbial succession and were designated as neonatally colonized ($LF82_N$) mice (Fig 1). Pups were weaned after 3 weeks of age and dams were removed from the isolators and placed into sterile cages on a HEPA-filtered air unit (e.g., Innovive) for the duration of the study. A separate subset of six week old ASF mice were colonized with *E. coli* LF82 by oral gavage and designated as adult colonized ($LF82_A$) mice. In order to evaluate the impact of *E. coli* LF82 as a bacterial provocateur, we purposefully administered a dose of dextran sodium sulfate (DSS) that induce mild to no observable clinical or microscopic inflammation. At 9 weeks of age, mice were treated with filter sterilized 2% DSS

## Experimental Protocol

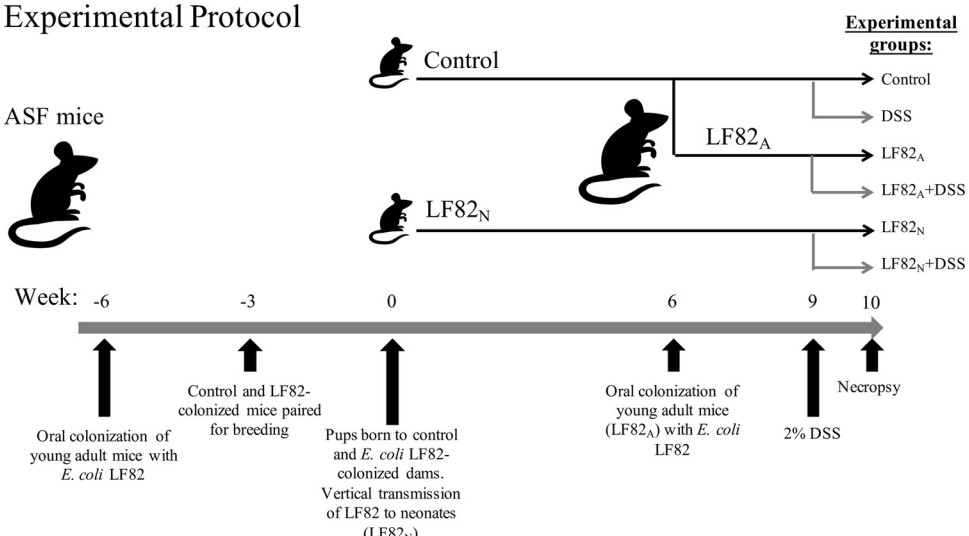

**Fig 1. Experimental protocol.** Adult gnotobiotic C3H/HeN mice harboring the altered Schaedler flora (ASF) were orally gavaged with *E. coli* LF82. After three weeks, LF82-infected and ASF-only control mice were paired for breeding. Pups born to *E. coli* LF82-infected dams were colonized by vertical transmission during the process of microbial succession and were designated as neonatal colonized (LF82$_N$). LF82$_N$ and ASF-only pups were weaned after 3 weeks of age. At 6 weeks of age, a subset of ASF-only mice were colonized with *E. coli* LF82 by oral gavage and designated as adult colonized (LF82$_A$). At 9 weeks of age, mice were treated with filter sterilized 2% dextran sodium sulfate in their drinking water for seven days prior to necropsy.

(MP Biomedicals, LLC, Santa Ana CA) in their drinking water for seven days after which time the mice were euthanized and necropsied. Fecal samples were aseptically collected prior to euthanasia and samples stored at -20°C until DNA extraction. Euthanasia was performed by $CO_2$ asphyxiation followed by exsanguination by cardiac puncture, and colonic tissues were harvested and cecal contents cultured to confirm the absence of aerobic contaminants. This study design was repeated four times, with the following additional group variations included in the later repeats. For neonatal horizontally colonized mice (LF82$_{NH}$), on day 0 (less than 24 hours of age) and again on day 2 after birth, pups were exposed to a suspension of $5 \times 10^8$ CFU/ml of *E. coli* LF82 by smearing the suspension on their face, body, and the dam's abdomen. A separate subset of *E. coli* LF82$_A$ mice had their period of colonization extended for six more weeks for a total of 10 weeks (10 wk LF82$_A$) before treatment with DSS as described above.

## Microbial colonization

Pathobiont *E. coli* LF82 was a kind gift from Kenny Simpson (Cornell, Ithaca, NY) [50]. Bacteria were streaked onto TSA plates and grown aerobically at 37°C. On the day of infection, bacteria were washed from plates and suspended in sterile PBS. The suspension was aseptically entered into the isolators or a laminar flow hood, and mice were infected with one dose (0.2 mL) of $5 \times 10^8$ CFUs by oral gavage. Confirmation of colonization was performed by bacteriological culture from fecal samples.

## Macroscopic and histopathological analysis

Excised cecum and colon were scored using a previously published scale for macroscopic inflammation [46]. Briefly, tissue was evaluated for macroscopic inflammatory changes of the cecum and colon including changes in tissue edema/thickening, presence of blood, the

presence or appearance of luminal contents, and general tissue morphology (e.g., cecal atrophy). Additionally, the length of the colon was measured.

Intestinal tissue was fixed in 10% neutral buffered formalin, paraffin embedded, sectioned, and stained with H&E. Tissue sections were evaluated by a board-certified veterinary pathologist (JH) blinded to the treatment groups as previously described [46]. Briefly, tissue was examined for epithelial damage, cellular infiltrate, and crypt/gland hyperplasia. Photomicrographs were taken on an Olympus Bx53 camera microscope with CellSense software.

## Cytokine analysis

Supernatants from incubated colonic tissue explants were harvested and prepared as previously described [46]. Supernatants were assayed for the presence of proinflammatory cytokines IL-17, IFNγ, IL-6, and IL-1β on the Bio-Plex 200 system (Bio-Rad, Hercules CA) using a multiplex bead-based assay. 2500 beads (Bio-Rad, Hercules CA) coupled to cytokine-specific monoclonal antibodies (eBioscience, San Diego CA) were added to each well along with 50 μL of culture supernatant. Cytokine standards (eBioscience, San Diego CA) were used to generate a standard curve covering cytokine concentrations from 500,000 to 3.2 pg/ml. The biotinylated detection antibodies (eBioscience, San Diego CA) were used at a final concentration of 2 to 4 μg/mL, and PE streptavidin (eBioscience, San Diego CA) was added for detection at 10 μg/mL.

## *E. coli* quantification

Cecal contents at necropsy were collected and serially diluted in sterile PBS and plated in triplicate on McConkey agar. After overnight incubation at 37°C, *E.* coli LF 82 CFUs were enumerated.

## 16s rRNA gene amplicon sequencing

Genomic DNA (gDNA) isolation from fecal samples was performed using MoBio's PowerSoil-htp DNA Isolation Kit (MoBio, Vancouver BC)(now QIAGEN DNeasy PowerSoil Pro) as recommended by the manufacturer. DNA was quantified with a Qubit 3.0 Fluorometer (Life Technologies, Carlsbad CA) and stored at -20°C in the supplied 10 mM TRIS buffer. DNA was sent to Argonne National Laboratory's Institute for Genomics & Systems Biology for PCR amplification of the V4 variable region of the 16S rRNA gene using V4 region specific primers (515F-816R) and amplicon sequencing on the Illumina MiSeq Platform. QIIME (Quantitative Insights into Microbial Ecology) was used to analyze sequences [51]. Briefly, 150 bp single end reads were demultiplexed and quality filtered to remove sequences with homopolymer runs or ambiguous bases greater than 6, nonmatching barcodes, barcode errors, or quality scores less than 25. A custom designed database, including only the 16S rRNA sequences of the ASF organisms, challenge organisms, and likely contaminants, was used to call operational taxonomic units (OTUs) at 97% similarity with uclust and the closed reference OTUs in QIIME [52]. No other organisms were detected besides the ASF and *E. coli*. Weighted UniFrac PCoA plots were generated using phyloseq in R (R Project) [53, 54]. Kruskal-Wallis tests were coupled with Wilcoxon Rank Sum tests and performed on taxonomic summaries, obtained from the QIIME pipeline, using a custom R script developed by the Institute for Genome Sciences at the University of Maryland School of Medicine.

## Fluorescent *in situ* hybridization

Tissues were processed, hybridized with FISH probes, stained with 4,6-diamidino-2-phenylindole (DAPI), imaged, and counted as previously described [55]. The Eubacteria probe

(Eub338) has been previous described, and the *E. coli* probe sequence was `5′-GCAAAGG-TATTAACTTTACTCCC-3′` [55]. Briefly, sections of proximal colon with intact contents were fixed overnight in formalin before moving to 70% ethanol and paraffin embedded. 3-μM thick sections were deparaffinized, labeled with probes overnight, and stained with DAPI. Tissues were imaged with x60 Plan Apo oil objective in conjunction with an optional x1.5 multiplier lens on an Eclipse TE2000-E fluorescence microscope (Nikon Instruments Inc., Minato, Tokyo Japan). A CoolSnap EZ camera (Photometrics, Tuscon AZ) with MetaMorph software (Nashville, TN) was used to photograph images. Counting was performed as previously described with 8–10 sections of each tissue in the lumenal, attached, and invasion compartments [55].

## RNAscope® *in situ* hybridization

Proximal colonic tissue was fixed and paraffin embedded as described above for histopathologic assessment. RNAscope® *in situ* hybridization was performed as recommended by the manufacturer with the RNAscope® 2.5HD Duplex assay with probes targeting CD4 (C2, channel #2) and IFNγ or IL-17 (C1, channel #1) (Advanced Cell Diagnostics, Newark CA). Briefly, 5 μM sections of paraffin embedded tissue were deparaffinized, treated with hydrogen peroxide, steamed for target retrieval, and treated with protease to prepare the samples. Next, the tissues were hybridized at 40˚C and repeatedly washed and hybridized for amplification sequentially for each probe before being stained with hematoxylin, dried, and mounted on glass slides. RNAscope® images were taken on an Olympus Bx53 camera microscope with CellSense software (Olympus, Shinjuku, Tokyo Japan). Halo® image analysis platform (Indica Labs, Albuquerque, NM) was used to quantitate RNA-labeling within the lamina propria and epithelial compartments separately. Five images captured at 60x from three tissues each were examined for RNA-labeling, and the five replicates were averaged for each tissue.

## Statistics

Prism 6 software was used for initial statistical analysis (GraphPad, La Jolla CA). A one way ANOVA with Tukey's multiple comparisons test or a Kruskal-Wallis is Dunn's multiple comparisons test were first used for parametric and nonparametric data, respectively. For the disease assessment indices, if the initial test demonstrated no statistical difference between $LF82_A$ + DSS and $LF82_N$ + DSS (or other groups in question), an unpaired t test or a two-tailed Mann-Whitney analysis was performed to directly compare the groups in question. Data has been uploaded to the data repository OSF (Open Science Format, DOI 10.17605/OSF.IO/9KVJR).

In addition, a simple linear regression model with an interaction term was created to analyze colon lengths versus cytokine secretion (IL-17, IFNγ, and IL-1β) from explants or versus histological scores. This model is described as $y_{ijk} = \alpha_i + \beta_j + \gamma_{ij} + e_{ijk}$ where $i = 1,2$ for DSS treatment, $i = 1$ is no DSS treatment, $i = 2$ is DSS treatment; $j = 1, 2, 3$ for groups, $j = 1$ is $LF82_A$, $j = 2$ is $LF82_N$, and $j = 3$ is control (no LF82); $k = 1,2,\ldots,n_{ij}$ are the individual observations in each group and treatment; $n_{11} = 21$, $n_{12} = 24$, $n_{13} = 17$; $n_{21} = 28$, $n_{22} = 32$ and $n_{23} = 21$; $e_{ijk}$ are independent and identically distributed random variables following a normal distribution with mean 0 and unknown variance $\sigma^2$. Data was analyzed in SAS software 9.4 (Cary NC). Colon lengths were initially analyzed with all groups, and then re-analyzed with only the DSS groups in the model (S3 Fig).

PCA plots were generated using R and included colon length, histology score and macroscopic score, as well as those three parameters plus cytokine secretion (IL-17, IFNγ, and IL-1β)

from colonic explants. R code can be found at https://github.com/FriedbergLab/neonatal_mouse_microbiome.

## Results

### ASF mice vertically colonized with *E. coli* LF82 at birth develop macroscopic lesions and severe, transmural histologic inflammation following exposure to DSS

Gnotobiotic mice harboring the ASF were colonized with *E. coli* LF82 by oral gavage as an adult (LF82$_A$) or by vertical acquisition as a neonate (LF82$_N$) to evaluate differences in the severity of DSS-induced colitis as influenced by the age at which the mice were colonized with *E. coli* LF82. DSS-induced inflammation decreased colon length and increased macroscopic and histologic scores (Fig 2). Contrary to our hypothesis, LF82$_N$ + DSS mice developed more severe macroscopic lesions (Fig 2A & 2B) with histologic inflammation (Fig 2C) when compared to all other groups of mice including LF82$_A$ + DSS mice. When using a linear regression statistical model with an interaction term (see methods), colon length was not significantly different with $p = 0.4325$ when all six groups were considered in the statistical model. When utilizing the linear regression model to selectively analyze across the three DSS treatment groups,

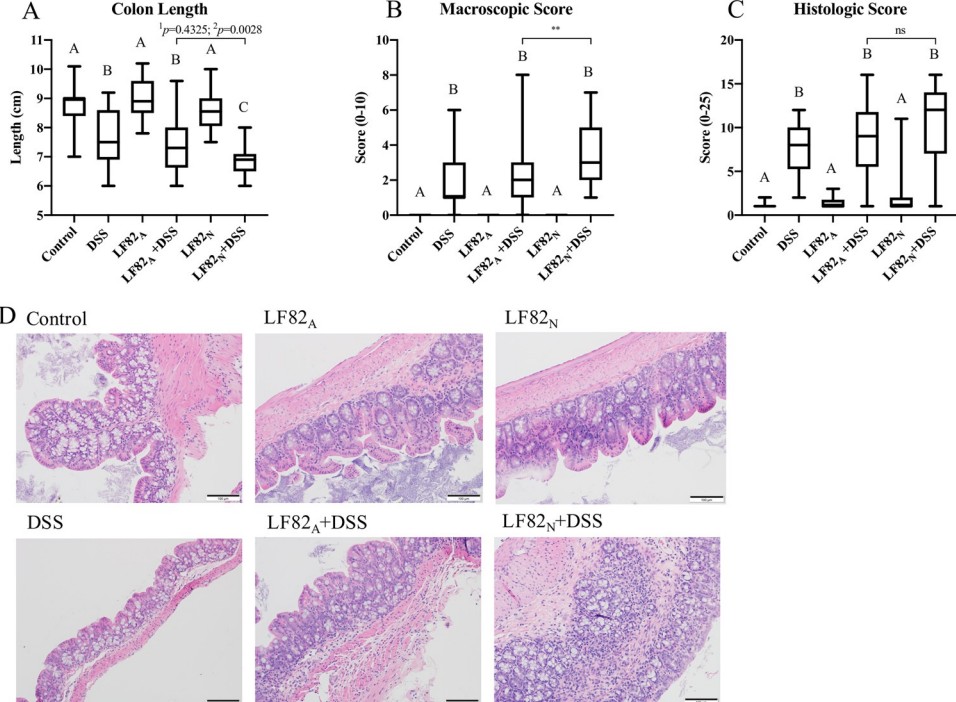

**Fig 2.** *E. coli* **LF82$_N$ mice develop severe, transmural colitis following DSS-induced inflammation.** Mice were scored at the time of necropsy for macroscopic parameters of disease including colon length (**A**) and macroscopic colitis score (as described in Materials and Methods) (**B**). Additionally, proximal colon was blindly scored for histopathologic parameters of inflammation by a pathologist (JH) (**C**). Representative photomicrographs of histopathological sections imaged at 40X (scale bar = 100 μm) (**D**). Shared letters between different groups indicate no statistical significance with an ANOVA or Kruskal-Wallis with multiple group comparisons test. Asterisk indicate statistical significance based on a direct comparison with a Mann-Whitney two-tailed test, $^*p \leq 0.05$, $^{**}p \leq 0.01$. $p$ for colon length data indicates significance based on a linear regression model for [1]all groups and between the [2]DSS treated groups. A and B: Control n = 17, DSS n = 21, LF82$_A$ n = 21, LF82$_A$ + DSS n = 28, LF82$_N$ n = 24, LF82$_N$ + DSS n = 32; C: Control n = 13, DSS n = 12, LF82$_A$ n = 16, LF82$_A$ + DSS n = 24, LF82$_N$ n = 19, LF82$_N$ + DSS n = 29.

the colon length was significantly different between LF82$_N$ + DSS and LF82$_A$ + DSS ($p$ = 0.0028).

While the difference in total histologic scores between LF82$_A$ + DSS and LF82$_N$ + DSS was not significant by ANOVA or a direct comparison with a Mann-Whitney test, there were higher scores in LF82$_N$ + DSS treatment group. Unlike the erosive lesions induced by DSS only, the lesions in the LF82$_N$ + DSS mice were characterized by marked multifocal mucosal ulceration and stromal collapse with a pronounced mixed inflammatory cell infiltrate that extended through the tunica muscularis (i.e., transmural in nature) of the colon. Hyperplasia of colonic glands was present over areas of intact epithelium (Fig 2D).

Horizontal colonization of mice at 24 hours of age with *E. coli* LF82 (LF82$_{NH}$), as opposed to vertical transmission from the dam, was also investigated. LF82$_{NH}$ mice treated with DSS presented with colon lengths, gross scores, and histologic scores that were intermediate between LF82$_A$ + DSS and LF82$_N$ + DSS mice (S1 Fig) suggesting that there was a maternal influence on disease severity. To address the duration of time that the mice were colonized with *E. coli* LF82, the severity of colitis in LF82$_A$ mice was evaluated after colonization for 10 weeks to be equivalent to the period of time that the LF82$_N$ mice were colonized prior to DSS exposure. The results indicate that the extending the post-colonization time period after *E. coli* LF82 colonization in adult mice did not increase the severity of DSS-induced colitis (S1F & S1G Fig).

## Minimal changes in the microbial community following *E. coli* LF82 colonization and/or DSS-induced inflammation

Based on bacteriological culture of fecal and cecal contents, *E. coli* LF82 readily and persistently colonized both LF82$_A$ and LF82$_N$ mice, and the numbers of *E. coli* LF82 in the cecal contents of mice increased approximately 3- to 5-fold following induction of DSS-induced inflammation (Fig 3A). Based on 16s rRNA taxonomic profiling, neither *E. coli* LF82 colonization nor DSS treatment significantly altered the relative abundance of the microbial community following multi-group statistical analyses (Fig 3B). Direct comparison of LF82$_A$ + DSS to LF82$_N$ + DSS and LF82$_{NH}$ + DSS to LF82$_N$ + DSS also revealed no statistically significant differences in the composition of the ASF community. Visualization of the community structures by PCoA analysis revealed that most of the treatment groups clustered closely together, while LF82$_A$ + DSS demonstrated more variability with a slight shift in microbial abundance, while LF82$_N$ + DSS demonstrated a more marked drift in microbial composition away from the rest of the groups (Fig 3C).

To assess whether microbial association with the colon or epithelial invasion may have contributed to the increased sensitivity to DSS, tissue sections of proximal colons with intact contents from control, LF82$_A$ + DSS, and LF82$_N$ + DSS mice were assessed for microbial spatial changes by fluorescent in situ hybridization (FISH). No significant changes were observed in total bacterial distribution in the luminal, adherent/attaching, or invading compartments (Fig 4A). In particular, three color FISH analysis demonstrated that *E. coli* LF82 could be detected within the luminal contents and there was no evidence that *E. coli* LF82 became more closely associated with the mucosa or invaded the colonic glands in LF82$_N$ + DSS mice (Fig 4B).

## Severity of colitis associated with differential production of proinflammatory cytokines

For the tissue samples collected from the mice in the LF82$_N$ + DSS treatment group on day seven of the DSS exposure, significantly more IL-17 was secreted from colonic explants when compared to explants from LF82$_A$ + DSS mice or mice treated with DSS alone (Fig 5A). In

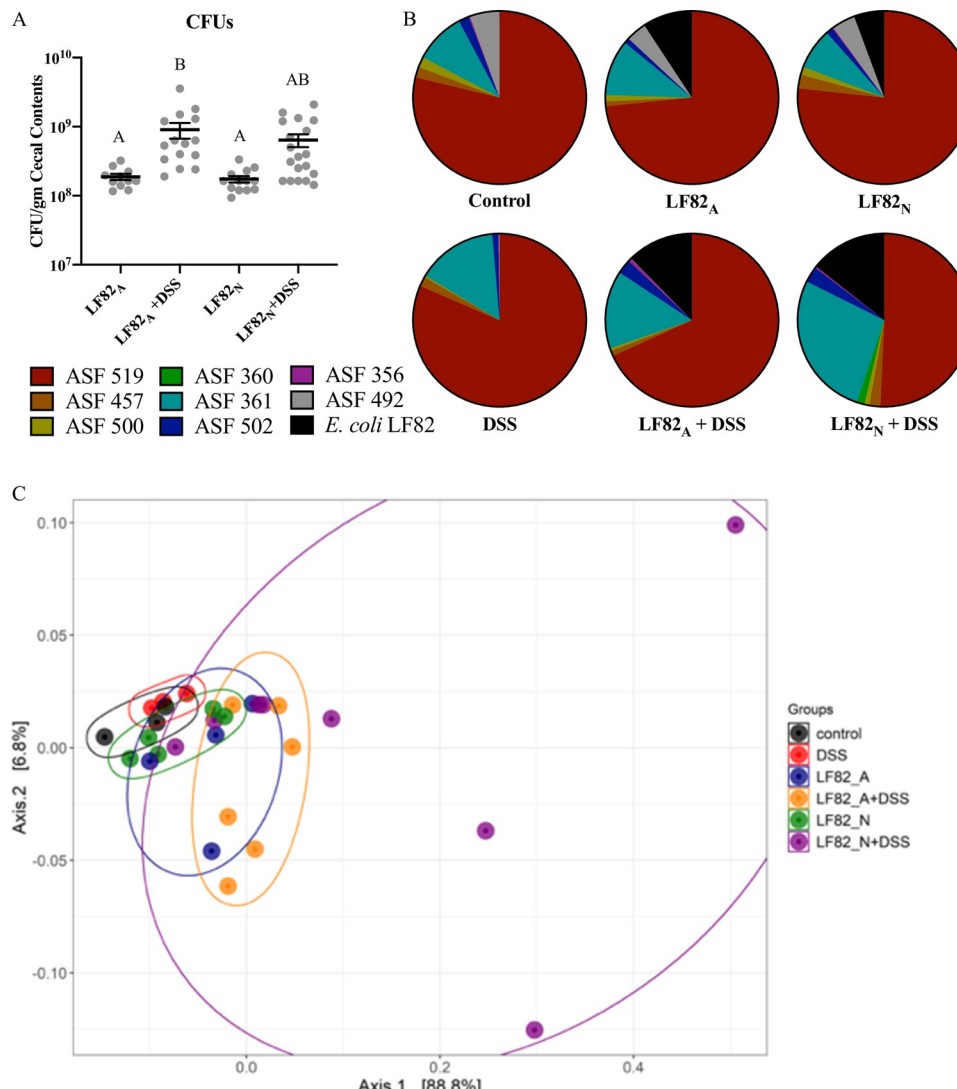

**Fig 3. Evaluation of the luminal microbial community following *E. coli* LF82 colonization and/or DSS-induced inflammation.** At necropsy, the cecal load of *E. coli* LF82 was enumerated by bacteriological culture (**A**), and 16s rRNA gene amplicon sequencing was performed on DNA recovered from fecal samples to assess relative microbial abundance (**B**). A Weighted UniFrac PCoA plot was generated using PhyloSeq to assess changes in beta diversity among the treatment groups (**C**). Each point represents a single animal, and each color represents a treatment group. The ellipses were generated to visualize the relative variability of each treatment group. Shared letters between different groups indicate no statistical significance with an ANOVA with multiple group comparisons test. A: LF82$_A$ n = 11, LF82$_A$ + DSS n = 15, LF82$_N$ n = 13, LF82$_N$ + DSS n = 19; B and C: Control n = 3, DSS n = 3, LF82$_A$ n = 4, LF82$_A$ + DSS n = 6, LF82$_N$ n = 5, LF82$_N$ + DSS n = 8.

contrast, IFNγ secretion was elevated in both LF82$_N$ + DSS and LF82$_A$ + DSS mice compared to mice treated with DSS alone (Fig 5B). Additionally, there was more IL-1β secreted from the mucosa of LF82$_N$ + DSS mice but it was not statistically different; the amount of IL-6 that was released was similar in all DSS treated groups (Fig 5C and 5D).

Consistent with the explant data, examination of the lamina propria for presence of IFNγ- and IL-17-specific mRNA by RNAscope® demonstrated that expression of both cytokines was elevated in the epithelium and lamina propria and generally co-localized with cells also

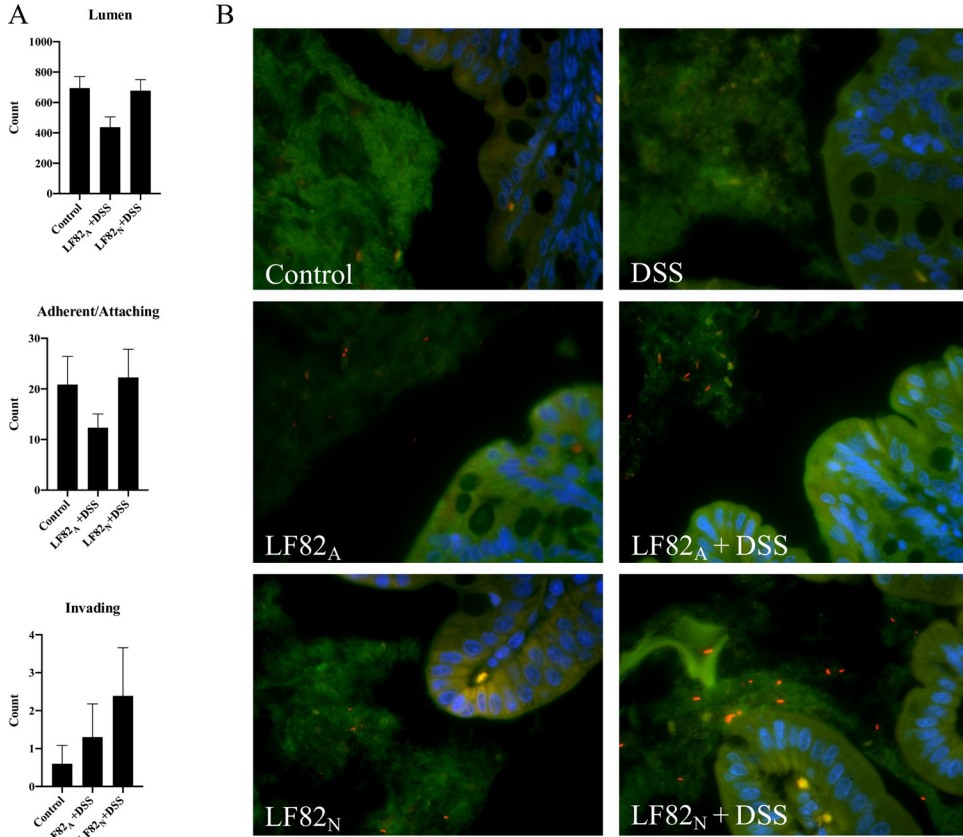

**Fig 4. Evaluation of spatial distribution of the microbial community following *E. coli* LF82 colonization and DSS-induced inflammation.** Proximal colons with intact contents were assessed for overall microbial spatial distribution by fluorescent in situ hybridization (FISH). Spatial distribution of organisms was evaluated using a eubacteria probe (EUB338) to detect bacteria in the lumen, adherent/attaching mucus layer, and translocation into the glands (**A**). Representative photomicrographs (60X magnification) of *E. coli* LF82 detected in the microbial community by FISH by labeling the nuclei of mucosal cells with DAPI (blue), total bacteria with a EUB338 probe (FITC, green), and labeling with a *E. coli*-specific probe (Cy-3, orange) (**B**). Control n = 5, LF82$_A$ + DSS n = 8, LF82$_N$ + DSS n = 8.

expressing CD4$^+$-mRNA (Table 1, S2 Fig). To quantify the number of cells expressing the specific mRNA, Halo$^\circledR$ software was utilized to count positive cells in the epithelial and lamina proprial compartments in LF82$_A$ + DSS and LF82$_N$ + DSS mice for presence of IFNγ- and IL-17-mRNA positive cells. Both the epithelium and the lamina propria of LF82$_N$ + DSS mice showed significantly increased numbers of IFNγ$^+$ cells and cells dual labeled for CD4- and IFNγ-specific mRNA compared to LF82$_A$ + DSS mice (Table 1). Additionally, both the epithelium and lamina propria demonstrated increases in the percent of IFNγ$^+$ cells that were also CD4$^+$, and indicated that for the LF82$_N$ + DSS mice the majority of the cells expressing IFNγ-specific mRNA also expressed CD4-specific mRNA; however, in the LF82$_A$ + DSS mice, most cells expressing IFNγ-specific mRNA did not co-express CD4-specific mRNA. In contrast, there was no significant difference in percent of IL-17 or cells dual labeled for CD4 and IL-17 expression in either compartment. Additionally with the exception of the epithelium in LF82$_A$ + DSS mice, most cells expressing IL-17-specific mRNA are negative for CD4-specific mRNA expression.

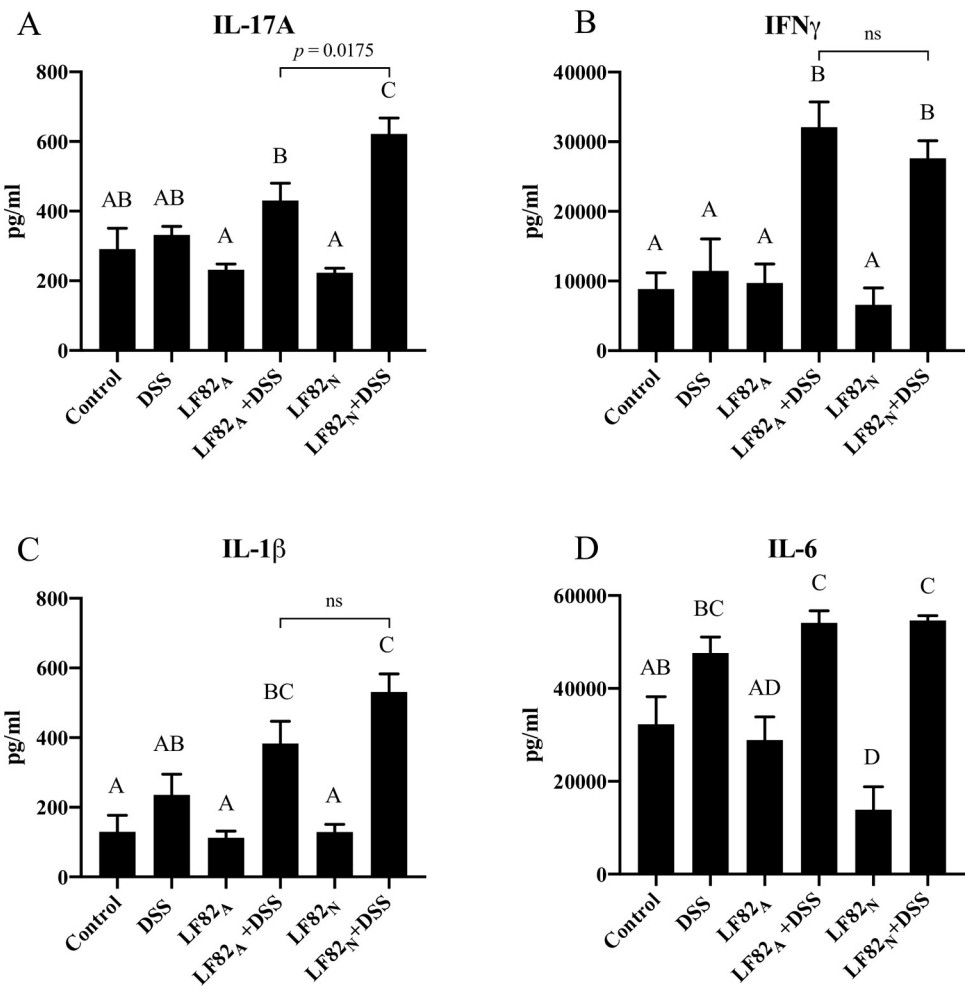

**Fig 5. Inflammation induces differential production of proinflammatory cytokines by colonic explants.** Analysis of IL-17A (**A**), IFNγ (**B**), IL-1β (**C**), and IL-6 (**D**) released from ex vivo incubated colonic explants. Mice were euthanized and tissues collected on day seven of the DSS exposure. Shared letters between different groups indicate no statistical significance based on an ANOVA with multiple group comparisons test. $p$ indicates significance based on a linear regression statistical model of the IL-17A secretion. Data is presented as the mean ± SEM. A-C: Control n = 11, DSS n = 9, LF82$_A$ n = 14, LF82$_A$ + DSS n = 19, LF82$_N$ n = 15, LF82$_N$ + DSS n = 24; D: Control n = 6, DSS n = 7, LF82$_A$ n = 6, LF82$_A$ + DSS n = 12, LF82$_N$ n = 8, LF82$_N$ + DSS n = 12.

## PCA demonstrates variation with inflammation

To assess the relative differences between each treatment group, a composite PCA plot was generated that included colon length, macroscopic score, and histological score (Fig 6A). The PCA plot demonstrated variation between the DSS treated groups in that the LF82$_N$ + DSS group did not cluster with the DSS only or LF82$_A$ + DSS treatment groups. An additional composite PCA plot was generated with the addition of tissue cytokine production (Fig 6B). Using these parameters, this analysis showed that *E. coli* LF82 colonization alone (i.e., neonatal or adult) did not differentiate these mice from the controls (black symbols). In the context of inflammation (i.e., after exposure to DSS, red symbols), the LF82$_N$ + DSS mice separated from the DSS only treatment group and the LF82$_A$ + DSS treatment group supporting the observation that the LF82$_N$ + DSS mice developed more severe lesions based on the collective responses assessed rather than any one measure of inflammation.

**Table 1. RNAscope® analysis with HALO® image analysis software.**

| | | IFNγ | | | IL-17 | | |
|---|---|---|---|---|---|---|---|
| | | %IFNγ⁺ | %Dual Positive | ᵃ%IFNγ⁺/CD4⁺ | %IL-17⁺ | %Dual Positive | ᵇ%IL-17⁺/CD4⁺ |
| LF82_A+DSS | epithelium | 0.3 | 0.157 | 24.56 | 0.78 | 0.60 | 62.69 |
| | | ± 0.10 | ± 0.06 | ± 3.75 | ± 0.61 | ± 0.46 | ± 11.05 |
| | lamina propria | 0.62 | 0.21 | 21.73 | 1.22 | 0.39 | 34.27 |
| | | ± 0.28 | ± 0.09 | ± 7.37 | ± 0.62 | ± 0.15 | ± 11.53 |
| LF82_N+DSS | epithelium | 4.77* | 2.79* | 60.42** | 0.65 | 0.35 | 39.17 |
| | | ± 1.35 | ± 0.87 | ± 2.16 | ± 0.38 | ±0.20 | ± 7.69 |
| | lamina propria | 2.8* | 1.92** | 60.37* | 0.45 | 0.17 | 44.44 |
| | | ± 0.60 | ± 0.35 | ± 10.16 | ± 0.12 | ± 0.01 | ± 10.78 |

Data is percent of total cells in that compartment ± standard error of the mean

ᵃPercent of IFNγ⁺ cells also CD4⁺

ᵇPercent of IL-17⁺ cells also CD4⁺

*$p \leq 0.05$

**$p \leq 0.01$

## Discussion

While IBD is often a disease diagnosed in the second or later decades of life, epidemiologic evidence indicates that IBD is increasingly being diagnosed in pediatric patients [35, 41]. IBD has long been associated with increased immunoreactivity against the resident microbiota, which has been confirmed in numerous experimental models [35, 56–59]. Bacterial pathobionts, such as AIEC, have been implicated in the pathogenesis of IBD where these organisms, that may colonize well after birth, act as a provocateur and perturb microbial composition and disrupt mucosal homeostasis [37, 45, 60–62]. Other members of the gastrointestinal microbiota, such as *Achromobacter pulmonis*, *Proteus mirabilis*, *Klebsiella pneumoniae*, and *Akkermansia muciniphila*, have also been shown to be pathobionts [63–65]. It has been shown that the hemolysin secreted from *P. mirabilis* can predispose to more severe colitis by activating NLRP3 and IL-1ß secretion while ability of *A. muciniphila* to degrade intestinal mucin facilitates the activation of NLRP6 in genetically susceptible individuals [64, 65]. He et al (2021)

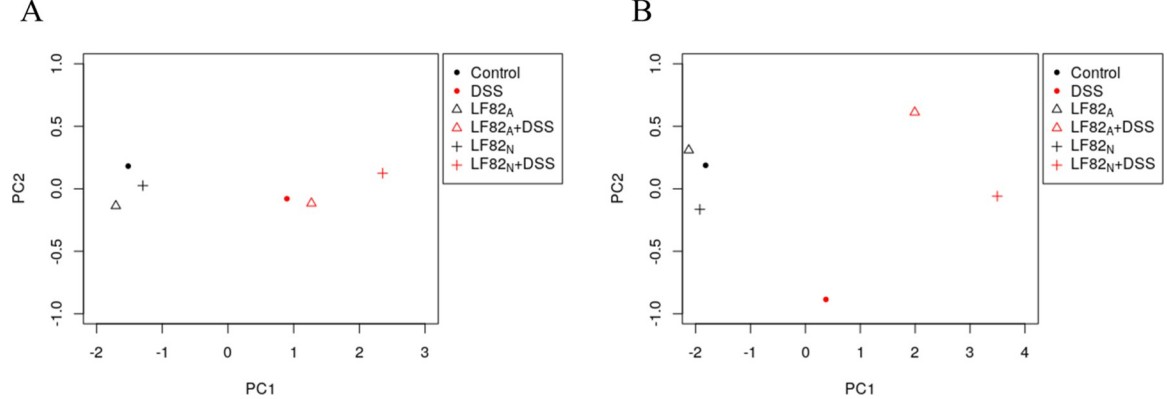

**Fig 6. PCA plots.** PCA plot depicting the clustering of samples originating from control ASF mice and ASF mice colonized with *E. coli* LF82 in the presence or absence of DSS exposure. A) PCA plot generated by incorporating data based on measurements of colon length, macroscopic lesions, and histological evaluation and B) PCA plot generated by incorporating data based on measurements of colon length, macroscopic, and histologic scores plus tissue cytokine production.

demonstrated the ability of *A. pulmonis* to exacerbate colitis was associated with its ability to impair the colonic mucus layer and to colonize mesenteric fat [66]. Therefore, the mechanisms by which a member of the intestinal microbiota function as a pathobiont are varied and likely contextual. In this study, it was hypothesized that mice colonized with *E. coli* LF82 as a neonate would have less severe DSS-induced colitis when compared to mice colonized with *E. coli* LF82 as an adult. Given that provocateurs and pathobionts are infrequently overt pathogens, our rationale was that colonization of neonates with a microorganism normally considered a provocateur would induce host mucosal tolerance or immunity, and the microorganism would be assimilated into the resident microbiota reducing its impact as a provocateur [46, 57–59]. Contrary to our hypothesis, the results of this study demonstrated that there was increased severity of DSS-induced colitis in mice that were vertically colonized with *E. coli* LF82 as a neonate (LF82$_N$) compared to being colonized as a young adult (LF82$_A$). Therefore, the concept of "Biological Freudianism" put forth by Dubos and colleagues appears to contribute to the observations described herein [21, 22].

The colitis observed in the LF82$_N$ + DSS mice was characterized by decreased colon lengths and increased histologic and macroscopic scores in comparison to mice in the LF82$_A$ + DSS treatment group. Histopathologically, many of the LF82$_N$ + DSS mice presented with transmural colonic lesions (Fig 2D) that morphologically resemble those observed in Crohn's disease patients as opposed to the ulcerative colitis-like lesions observed in the mice treated with DSS only [67, 68]. Additionally, neonatal mice borne to naïve dams that were infected horizontally with *E. coli* LF82 (LF82$_{NH}$) exhibited intermediate lesion severity when compared to LF82$_A$ and LF82$_N$ mice following DSS-induced inflammation (S1 Fig). Together, these observations suggest a complex disease etiology involving not only the presence of the pathobiont but also the dam. Such complex etiology involving a maternal effect is consistent with the classic dogma of IBD being due to a complicated, interacting mechanism involving the microbiome, genetic susceptibility, immune response, and environmental triggers (i.e., which may include the dam) [12, 35]. Given that the *E. coli* LF82$_N$ mice were colonized for approximately 10 weeks prior to exposure to DSS, a separate cohort of adult mice were colonized with *E. coli* LF82 for 10 weeks prior to DSS exposure (S1F & S1G Fig). Increasing the time during which the adult mice were colonized with *E. coli* LF82 did not result in the induction of more severe DSS-induced colitis in these LF82$_A$ + DSS mice.

It has previously been demonstrated that *E. coli* LF82 cannot colonize or cause disease in immunocompetent, conventionally-reared mice, unless transgenic for the human CEACAM 6 receptor, are immunodeficient, or are pretreated with antibiotics [42–45, 61, 62, 69]. An advantage of the limited microbiome of the ASF model is that colonization with proteobacteria (e.g., multiple strains of *Campylobacter jejuni* and *E. coli*) that will not readily colonize genetically wild-type, conventionally-reared mice is readily achieved [46, 70–72]. In this study, there were no differences noted in the composition of the ASF community based on 16s rRNA gene amplicon sequence analysis. However, the microbiota of mice infected with *E. coli* LF82 as a neonate presented with a change in the relative abundance of the microbial community composition compared to the rest of the treatment groups (e.g., control or LF82$_A$) (Fig 3C). In addition, FISH analysis of the ASF community demonstrated no significant spatial changes in distribution of the mucosal microbiota. Similarly, a recent study from another group with genetically wildtype, gnotobiotic ASF mice likewise noted minimal changes in the ASF community following the colonization of *E. coli* LF82 [48]. In contrast, mice that are transgenic for the expression of CEACAM6, where *E. coli* LF82 can attach to the mucosa, demonstrated chronically altered fecal microbial composition following LF82 colonization [73]. As anticipated, FISH demonstrated that *E. coli* LF82 is not adherent or invasive in ASF C3H/HeN mice, with only a few organisms becoming associated with the mucosa after DSS-induced

inflammation. Thus, the mechanisms associated with enhanced disease in the neonates are unrelated to the adherent and invasive capabilities of AIEC suggesting that the consequences of host-microbe interactions go beyond classical virulence traits.

Cytokines are important in intestinal homeostasis and inflammation since they can perpetuate or control the inflammatory response. Increases in many proinflammatory cytokines are associated with IBD, and multiple cytokine neutralizing therapeutics (biologics) are currently in use or clinical trials for treatment of IBD and other chronic inflammatory conditions [74]. Colitis in the dual treated mice (i.e., LF82 + DSS) was accompanied by elevated proinflammatory cytokines in the colonic tissues including IL-17, IFNγ, IL-1β, and IL-6 compared to the mice treated with DSS only (Fig 5). In this regard, the LF82$_N$ + DSS mice were observed to have secreted significantly increased levels of IL-17 and a trend for increased IL-1β as compared to the mice in the LF82$_A$ + DSS treatment group. Both IL-17 and IL-1ß have all been implicated in the pathogenesis of IBD [74, 75]. Depending on the context of the inflammation, IL-17 has a complicated role in IBD, with protective and/or pathogenic effects as evidenced by a recent study suggesting that IL-17 has a protective effect in the DSS- and AIEC-induced colitis models [76, 77].

RNAscope® was utilized to evaluate *in situ* cytokine production in the colon, with a limited objective assessment performed by computer software to determine location and to associate mRNA expression to specific immune cells within the lamina propria and epithelium. An increase in cells expressing IFNγ-specific mRNA was demonstrated in the epithelium and lamina propria of the LF82$_N$ + DSS treatment group as compared to the LF82$_A$ + DSS treated mice, in contrast to the colonic explants where there was no difference in IFNγ secretion between these groups. Additionally, in contrast to the explant cytokine analysis, there was no difference in the number of CD4$^+$ cells expressing IL-17-specific mRNA. This may be explained in that RNAscope® is measuring RNA transcript while the explant cytokine analysis is measuring protein levels. Additionally, RNAscope measurements are reported on a per cell basis as opposed to total cytokine secretion from an equally-sized piece of tissue, and there may be increased production by the same cells rather than an increase in cells producing cytokines or cells expressing IL-17-specific mRNA. A majority of the cells expressing IFNγ-specific mRNA in LF82$_N$ + DSS mice were also positive for CD4-specific mRNA (~60%), though a smaller percentage of IL-17$^+$ cells were also positive for CD4-specific mRNA (~42%) suggesting that most IL-17 secreting cells are not CD4$^+$ T cells (~58%), and thus other cell types (e.g., ILC3, NKT) are contributing to the production of these cytokines. Regardless, both the RNAscope and explant cytokine analysis identify differential patterns of cytokine production between the LF82$_N$ + DSS and LF82$_A$ + DSS treatment groups.

The concept that early microbial exposures and environmental factors (e.g., Biological Freudianism or biological imprinting) have a lasting influence on the host has been studied since at least the 1960s [21, 22]. Since then, more studies have demonstrated these early exposures can have a lasting effect on disease development, especially in relation chronic inflammatory conditions like IBD, obesity, and asthma [30–34]. A research study using reversible colonization in pregnant mice, demonstrated that the dam's microbial status does impact the developing neonatal immune system, with differential immune cell development, altered intestinal gene expression, and increased protection from bacterial translocation in the offspring [13]. A study evaluating neonatal colonic inflammation in rats demonstrated that the increased inflammatory responses observed in adult animals was due to epigenetic changes of the IL-1β promoter that resulted in increased expression following a colitic insult later in life [78]. A recent paper demonstrated that transient infection of the maternal gastrointestinal tract could alter the mucosal immune response of offspring that was associated with epigenetic changes regulated by IL-6 production that contributed to expanded Th17 cell populations in the gut of

the offspring which enhanced protective immunity to subsequent *Salmonella* infection [12]. In this same study, IL-6 originating in the dam contributed to expansion of Th17 cell populations in the gut of the offspring that were responsible for the enhanced resistance to subsequent bacterial infection but also enhanced sensitivity to colitis in the offspring later in life [12]. Epigenetic regulated mechanisms of gene expression have been evaluated in association with the pathogenesis of IBD [79]. While epigenetic analysis was beyond the scope of this work, it may be one mechanism contributing to the increased severity of DSS-induced colitis in neonatal mice colonized vertically from the infected dam.

In conclusion, the results of this study demonstrate that vertically acquired *E. coli* LF82 by neonatal ASF mice (LF82$_N$) resulted in more severe DSS-induced colitis as compared to adult mice colonized with *E. coli* LF82. Regardless of their age, the colonization of the gnotobiotic mice with *E. coli* LF82 minimally impacted the microbial community structure as measured by relative abundance or spatial redistribution. However, subsequent to a colitic trigger, an altered immune response was observed that was characterized by increased proinflammatory cytokines in the mucosal tissues. These data suggest that the age at which an individual is colonized by a pathobiont or provocateur may differentially affect the host's mucosal immune profile and impact the host's susceptibility to subsequent inflammatory insults later in life.

## Supporting information

**S1 Fig. LF82$_{NH}$ LF82 mice have intermediate colitis scores compared to LF82$_N$ and LF82$_A$ following DSS-induced inflammation.** Mice were horizontally infected as neonates (LF82$_{NH}$) to assess maternal influence on the effects observed following colonization with *E. coli* LF82. Mice were evaluated at the time of necropsy for parameters of disease including colon length (A) and macroscopic colitis score (B). Additionally, proximal colon was fixed, sectioned and stained with hematoxylin and eosin and blindly scored by a pathologist for histopathologic parameters of inflammation (C). The cecal load of *E. coli* LF82 was enumerated by bacteriological culture (D) and 16s rRNA gene amplicon sequencing was performed on DNA recovered from fecal samples to assess relative microbial abundance (E). To mimic the length of time that the neonates were colonized with *E. coli* LF82, a separate set of adult mice evaluated at 10 weeks post-colonization (10 wk LF82$_A$). Mice were scored at the time of necropsy for parameters of disease including colon length (F) and macroscopic colitis score (G). Shared letters between different groups indicate no statistical significance with an ANOVA or Kruskal-Wallis with multiple group comparisons test; ns indicates no statistical significance based on a direct comparison with a t or Mann-Whitney test. A and B: Control n = 17, DSS n = 21, LF82$_A$ n = 21, LF82$_A$ + DSS n = 28, LF82$_{NH}$ n = 15, LF82$_{NH}$ + DSS n = 18, LF82$_N$ n = 24, LF82$_N$ + DSS n = 32; C: Control n = 13, DSS n = 12, LF82$_A$ n = 16, LF82$_A$ + DSS n = 24, LF82$_{NH}$ n = 5, LF82$_{NH}$ + DSS n = 5, LF82$_N$ n = 19, LF82$_N$ + DSS n = 29; D and E: LF82$_{NH}$ n = 6, LF82$_{NH}$ + DSS n = 8; A and B: Control n = 17, DSS n = 21, LF82$_A$ n = 21, LF82$_A$ + DSS n = 28, 10 wk LF82$_A$ n = 4, 10 wk LF82$_A$ + DSS n = 5, LF82$_N$ n = 24, LF82$_N$ + DSS n = 32. (TIFF)

**S2 Fig. RNAscope detection of IL-17, IFNγ and CD4 mRNA transcripts in the colon of *E. coli* LF82 colonized mice treated with DSS.** IL-17, IFNγ, and CD4 mRNA was labeled with a duplex RNAscope® assay. Representative photomicrographs demonstrating dual labeling of CD4- (red) and cytokine-specific (green) mRNA in multiple experimental groups (scale bar = 20 μm). (TIFF)

**S3 Fig. Raw SAS outputs from analyses as described in materials & methods.**
(PDF)

## Acknowledgments

The authors thank Mary Jane Long for technical assistance throughout these studies, Ross Darling, Jorrell Fredericks, and Danielle Wagner for assistance with necropsy and initial sample processing and Katarina Kohn for assistance with data entry. Additionally thank you to Angela Bryan, Jessica Elbert, and Curtis Mosher for assistance with FISH imaging and counting, and Rachel Phillips for the RNAscope® imaging and Halo® analysis.

## Author Contributions

**Conceptualization:** Meghan Wymore Brand, Gregory J. Phillips, Michael J. Wannemuehler.

**Formal analysis:** Meghan Wymore Brand, Naihui Zhou, Iddo Friedberg.

**Funding acquisition:** Meghan Wymore Brand, Albert E. Jergens, Gregory J. Phillips, Michael J. Wannemuehler.

**Investigation:** Meghan Wymore Brand, Alexandra L. Proctor, Jesse M. Hostetter, Albert E. Jergens.

**Methodology:** Meghan Wymore Brand, Michael J. Wannemuehler.

**Project administration:** Michael J. Wannemuehler.

**Software:** Alexandra L. Proctor, Naihui Zhou, Iddo Friedberg.

**Supervision:** Michael J. Wannemuehler.

**Visualization:** Meghan Wymore Brand, Alexandra L. Proctor, Jesse M. Hostetter.

**Writing – original draft:** Meghan Wymore Brand, Michael J. Wannemuehler.

**Writing – review & editing:** Meghan Wymore Brand, Alexandra L. Proctor, Iddo Friedberg, Albert E. Jergens, Gregory J. Phillips, Michael J. Wannemuehler.

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
