## [Decision Letter · Decision Letter 0]

23 Dec 2021

PONE-D-21-36309Vertical transmission of attaching and invasive E. coli from the dam to neonatal mice predisposes to more severe colitis following exposure to a colitic insult later in lifePLOS ONE

Dear Dr. Wannemuehler

Thank you for submitting your manuscript to PLOS ONE. After careful consideration, we feel that it has merit but does not fully meet PLOS ONE’s publication criteria as it currently stands. Therefore, we invite you to submit a revised version of the manuscript that addresses the points raised during the review process.

Although one reviewer appreciates the importance of the study, the other reviewer has critical concerns about the authors' experimental design and  interpretation of their results. Thus, I would like to ask the authors to respond to the comments raised by the reviewers as far as they can. 

We look forward to receiving your revised manuscript.

Kind regards,

Hiroyasu Nakano, M.D., Ph.D.

Academic Editor

PLOS ONE

Journal Requirements:

[The authors thank Mary Jane Long for technical assistance throughout these studies, Ross Darling, Jorrell Fredericks, and Danielle Wagner for assistance with necropsy and initial sample processing. Additionally thank you to Angela Bryan, Jessica Elbert, and Curtis Mosher for  assistance with FISH imaging and counting, Katarina Kohn for assistance with QIIME analyses, and Rachel Phillips for the RNAscope® imaging and Halo® analysis. The authors acknowledge support from the following organizations: Kenneth Rainin Foundation Innovator Award (AEJ, MJW, GJP - https://krfoundation.org/health/grants/innovator-awards/); the NIH NIGMS award R01GM099537-1A1 (GLP, MJW, AEJ, JMH http://www.nigms.nih.gov); NSF DBI 1551363 (IF http://www.nsf.dbi.gov). The funders had no role in study design, data collection and analysis, decision to publish, or preparation of the manuscript].

 [Keneth Rainin Foundation Innovator Award (AEJ, MJW, GJP - https://krfoundation.org/health/grants/innovator-awards/);  the NIH NIGMS award R01GM099537-1A1 (GLP, MJW, AEJ, JMH   http://www.nigms.nih.gov); NSF DBI 1551363 (IF http://www.nsf.dbi.gov). The funders had no role in study design, data collection and analysis, decision to publish, or preparation of the manuscript.”]

Reviewers' comments:

Reviewer's Responses to Questions

**Comments to the Author**

1. Is the manuscript technically sound, and do the data support the conclusions?

Reviewer #1: Yes

Reviewer #2: No

2. Has the statistical analysis been performed appropriately and rigorously? 

Reviewer #1: Yes

Reviewer #2: I Don't Know

3. Have the authors made all data underlying the findings in their manuscript fully available?

Reviewer #1: Yes

Reviewer #2: No

4. Is the manuscript presented in an intelligible fashion and written in standard English?

Reviewer #1: Yes

Reviewer #2: Yes

5. Review Comments to the Author

Reviewer #1: In this study, Wannemuehler‘s group tried to demonstrate that an early exposure to microbiota beneficially impacts mucosal immunity later in life. To achieve this goal, authors examined whether neonatal colonization with AIEC provides lasting protection from DSS-induced colitis in ASF-colonized gnotobiotic mice. Contrary to the speculation that AIEC-colonized mice as a neonate develop less severe colitis induced by DSS in drinking water as compared with adult-colonized mice, the former group showed slightly increased severity of pathology associated with DSS-induced colitis than the latter group. This study addresses an important biological concept that early microbial exposures regulate the susceptibility against IBD later in life using a well-designed experimental protocol. Most of the statements are supported by experimental evidence and the manuscript is overall coherently written. Therefore, this paper could be accepted for publication to PLOS ONE subject to several amendments as described below.

Comments:

1. In Figure 2, although difference between vertically colonized mice and horizontally colonized ones is statistically significant, severity between two groups of mice is marginal at best. Considered that authors have rigorously performed statistical analysis using large number of mice, which was respectable, this reviewer is afraid whether AIEC-colonization during neonate has little or no impact on the mucosal immunity in terms of the severity of DSS-induced colitis. This reviewer proposes that authors mention about a possibility of pathobionts other than AIEC have larger impact on gut homeostasis in the DISCUSSION section.

2. Authors clearly showed that vertical colonization with AIEC does not change the mucosal pathology in steady-state. Explain more carefully the possible mechanism associated with the enhanced colitis, including enhanced colon IL-17 mRNA level, in mice colonized as a neonate, please. Again, related to comment 1, a possibility of vertical colonization with AIEC do regulate gut homeostasis in some manner but having little impact on the pathology of DSS-induced colitis could not be excluded.

Reviewer #2: The interaction of the microbiome with the mucosal immune system in the gut plays an important role in priming immunity. However, it is not fully understood how the microbiome regulates immune homeostasis. The authors have previously reported that E. coli LF82 colonization increases the susceptibility of adult C3H/HeN mice harboring the altered Schaedler flora (ASF) to dextran sodium sulfate (DSS)-induced colitis. In this study, the authors examined the impact of the pathobiont colonization of neonates on the severity of colitis later in life. The authors colonized gnotobiotic mice harboring the ASF with E. coli LF82 by oral gavage as an adult (LF82A) or by vertical acquisition as a neonate (LF82N) and evaluated the differences in the severity of DSS-induced colitis. The authors concluded that the DSS-induced colitis was exacerbated in the LF82N group compared to the LF82A group. Although the authors’ hypothesis appears to be interesting, the difference between the two groups (LF82N vs. LF82A) appears to be marginal. Moreover, there are several critical concerns about the authors’ experimental conditions.

The followings are specific comments.

Major points;

1. The authors concluded that the DSS-induced colitis was exacerbated in the LF82N group compared to the LF82A group based on the length of the colon. Although the macroscopic scores were correlated with the length of the colon, histological scores of the colitis between the two groups were not different. The authors need to discuss these points or replace them with more representative ones to support the authors’ conclusion.

1. In Figure 2D, the colonic mucosa in DSS-treated mice seemed to be normal. In general, DSS treatment usually induces focal ulceration of the colonic mucosa along with massive infiltration of mononuclear cells, as observed in the colon of LF82N + DSS. Thus, it is unclear why DSS treatment did not induce such changes. The authors need to explain apparent inconsistent results.

1. In Figure 4, it is unclear whether the positive signals actually represent the existence of the indicated bacteria. Why were the positive signals not detected in the lumen of the colon of LF82N mice? Given that LF82N mice harbored E. coli in the colon, the signals of E. coli should be present in the lumen of the colon. The authors should show these results in a more convincing manner.

What do G0 and G1 mean?

1. In Figure 5, there was no significant difference in cytokine production between the control and the DSS-treated grossssssups at least the authors’ experimental conditions. Given that DSS treatment induces severe inflammation and upregulation of many cytokines, the results seem to be surprising. To further substantiate the authors’ findings, the authors need to check the expression of several cytokines of the colon of untreated and DSS-treated mice by qPCR.

6. PLOS authors have the option to publish the peer review history of their article (what does this mean?). If published, this will include your full peer review and any attached files.

Reviewer #1: No

Reviewer #2: No

---

## [Author Response · Author response to Decision Letter 0]

4 Feb 2022

We have responded to the reviewer's comments and attached a file specifically detailing those responses. There are two copies of the manuscript uploaded, one with track changes.

---

## [Decision Letter · Decision Letter 1]

4 Mar 2022

PONE-D-21-36309R1Vertical transmission of attaching and invasive E. coli from the dam to neonatal mice predisposes to more severe colitis following exposure to a colitic insult later in lifePLOS ONE

Dear Dr. Wannemuehler

Thank you for submitting your manuscript to PLOS ONE. After careful consideration, we feel that it has merit but does not fully meet PLOS ONE’s publication criteria as it currently stands. Therefore, we invite you to submit a revised version of the manuscript that addresses the points raised during the review process. While both reviewers have been satisfied with the authors' revision, one reviewer still has minor concern. Please respond to the comments raised by the reviewer 2.  

We look forward to receiving your revised manuscript.

Kind regards,

Hiroyasu Nakano, M.D., Ph.D.

Academic Editor

PLOS ONE

Journal Requirements:

Reviewers' comments:

Reviewer's Responses to Questions

**Comments to the Author**

1. If the authors have adequately addressed your comments raised in a previous round of review and you feel that this manuscript is now acceptable for publication, you may indicate that here to bypass the “Comments to the Author” section, enter your conflict of interest statement in the “Confidential to Editor” section, and submit your "Accept" recommendation.

Reviewer #1: All comments have been addressed

Reviewer #2: All comments have been addressed

2. Is the manuscript technically sound, and do the data support the conclusions?

Reviewer #1: Yes

Reviewer #2: Partly

3. Has the statistical analysis been performed appropriately and rigorously? 

Reviewer #1: Yes

Reviewer #2: I Don't Know

4. Have the authors made all data underlying the findings in their manuscript fully available?

Reviewer #1: Yes

Reviewer #2: Yes

5. Is the manuscript presented in an intelligible fashion and written in standard English?

Reviewer #1: Yes

Reviewer #2: (No Response)

6. Review Comments to the Author

Reviewer #1: Authors have adequately responded to this reviewer's comments. The manuscript is now ready for publication to PLOSONE.

Reviewer #2: The authors have responded to the reviewers’ comments.

However, Figure 4 is still unclear. A low magnification picture showing the lumen and tissue, and a high magnification picture showing infiltration should be presented in the figure. In addition, they need to add the scales in the pictures.

7. PLOS authors have the option to publish the peer review history of their article (what does this mean?). If published, this will include your full peer review and any attached files.

Reviewer #1: No

Reviewer #2: No

---

## [Author Response · Author response to Decision Letter 1]

9 Mar 2022

We have made edits to the manuscript in response to Reviewer #2 comments with respect to Figure 4

Reviewer #2 

Figure 4 is still unclear. A low magnification picture showing the lumen and tissue, and a high magnification picture showing infiltration should be presented in the figure. In addition, they need to add the scales in the pictures.

We appreciate the Reviewer’s comment and considered it carefully as we do want this manuscript to be clear for all readers. First, in consultation with our co-author, Dr. Jesse Hostetter (Chair, Dept. Veterinary Pathology, University Georgia), there is little need or value to adding a scale bar to each of the images in Figure 4. While it was listed in the Methods section, we had not indicated the magnification of the photomicrographs in the Figure legend. This has now been added. It addition, these images demonstrate that the pathobiont, E. coli LF82, did not adhere to nor invade the epithelium. These observations support the following statement in the Discussion (lines 505 - 507 ) - Thus, the mechanisms associated with enhanced disease in the neonates are unrelated to the adherent and invasive capabilities of AIEC suggesting that the consequences of host-microbe interactions go beyond classical virulence traits.

The Reviewer also asked that we provide a higher magnification of the luminal contents and epithelium showing E. coli LF82 attachment or invasion. We appreciate that the Reviewer had anticipated that there would/might be epithelial invasion of the tissue by E. coli LF82 given that its pathotype is ‘adherent and invasive E. coli’ (AIEC). As stated above and described in the manuscript, E. coli LF82 does not invade the epithelium of mice given that mice do not express CEACAM6. Providing a higher magnification would not provide any additional information as there was no invasion noted. Again, we have consulted with Dr. Hostetter and he agrees that a higher magnification inset would not add any additional information. That said, we revised the paragraph in the Results section to make it more clear as to what our goal was with respect to showing these photomicrographs. Because of the Reviewers comment, we feel these edits were warranted and improved the manuscript. Here is the edited paragraph (lines 355 - 362):

To assess whether microbial association with the colon or epithelial invasion may have contributed to the increased sensitivity to DSS, tissue sections of proximal colons with intact contents from control, LF82A + DSS, and LF82N + DSS mice were assessed for microbial spatial changes by fluorescent in situ hybridization (FISH). No significant changes were observed in total bacterial distribution in the luminal, adherent/attaching, or invading compartments (Fig 4A). In particular, three color FISH analysis demonstrated that E. coli LF82 could be detected within the luminal contents and there was no evidence that E. coli LF82 became more closely associated with the mucosa or invaded the colonic glands in LF82N + DSS mice (Fig 4B).

---

## [Decision Letter · Decision Letter 2]

14 Mar 2022

Vertical transmission of attaching and invasive E. coli from the dam to neonatal mice predisposes to more severe colitis following exposure to a colitic insult later in life

PONE-D-21-36309R2

Dear Dr. Wannemuehler

We’re pleased to inform you that your manuscript has been judged scientifically suitable for publication and will be formally accepted for publication once it meets all outstanding technical requirements.

Kind regards,

Hiroyasu Nakano, M.D., Ph.D.

Academic Editor

PLOS ONE

Additional Editor Comments (optional):

Reviewers' comments:

Reviewer's Responses to Questions

**Comments to the Author**

1. If the authors have adequately addressed your comments raised in a previous round of review and you feel that this manuscript is now acceptable for publication, you may indicate that here to bypass the “Comments to the Author” section, enter your conflict of interest statement in the “Confidential to Editor” section, and submit your "Accept" recommendation.

Reviewer #1: All comments have been addressed

Reviewer #2: All comments have been addressed

2. Is the manuscript technically sound, and do the data support the conclusions?

Reviewer #1: Yes

Reviewer #2: Yes

3. Has the statistical analysis been performed appropriately and rigorously? 

Reviewer #1: Yes

Reviewer #2: I Don't Know

4. Have the authors made all data underlying the findings in their manuscript fully available?

Reviewer #1: Yes

Reviewer #2: Yes

5. Is the manuscript presented in an intelligible fashion and written in standard English?

Reviewer #1: Yes

Reviewer #2: Yes

6. Review Comments to the Author

Reviewer #1: Authors have adequately responded to reviewers' requests. The revised version of manuscript satisfies the publication policy of PLOS ONE.

Reviewer #2: (No Response)

7. PLOS authors have the option to publish the peer review history of their article (what does this mean?). If published, this will include your full peer review and any attached files.

Reviewer #1: No

Reviewer #2: No

---

## [Editor Report · Acceptance letter]

22 Mar 2022

PONE-D-21-36309R2 

Vertical transmission of attaching and invasive *E. coli* from the dam to neonatal mice predisposes to more severe colitis following exposure to a colitic insult later in life 

Dear Dr. Wannemuehler:

I'm pleased to inform you that your manuscript has been deemed suitable for publication in PLOS ONE. Congratulations! Your manuscript is now with our production department. 

Kind regards, 

on behalf of

Professor Hiroyasu Nakano 

Academic Editor

PLOS ONE